# Knowledge and Practice of Childbearing Women in Saudi Arabia towards Folic Acid Supplement—Evidence from a Cross-Sectional Study

**DOI:** 10.3390/nu14020357

**Published:** 2022-01-14

**Authors:** Mohamed N. Al Arifi, Ali M. Alqahtani, Abdulaziz Naif Alotaibi, Salmeen D. Babelghaith, Abdulrahman Alwhaibi, Sary Alsanea, Sultan M. Alghadeer, Nasser M. Al-Arifi

**Affiliations:** 1Department of Clinical Pharmacy, College of Pharmacy, King Saud University, Riyadh 11451, Saudi Arabia; A.22.naif@gmail.com (A.N.A.); sbabelghaith@ksu.edu.sa (S.D.B.); aalwhaibi@KSU.EDU.SA (A.A.); salghadeer@ksu.edu.sa (S.M.A.); 2Department of Pharmacy Services, Security Forces Hospital, Riyadh 11481, Saudi Arabia; Ali.3baid@gmail.com; 3Department of Pharmacology and Toxicology, College of Pharmacy, King Saud University, Riyadh 11451, Saudi Arabia; Salsanea@ksu.edu.sa; 4College of Medicine, Almaarefa University, Riyadh 11597, Saudi Arabia; Nasser7ar@gmail.com

**Keywords:** knowledge, practices, folic acid, birth defects, supplementation, Saudi Arabia

## Abstract

Background and objectives: Neural tube defects are congenital anomalies which canlead to infant death and serious disability. They are initiated during embryogenesis, between the 23rd and 27th day of fetal life, and can be prevented by the administration of folic acid. Therefore, this study aims to evaluate the knowledge and practice of Saudi women at childbearing age regarding NTDs and FA supplementation. Methodology: This is a cross-sectional study on Saudi women of reproductive age who were asked to complete an online survey to examine their knowledge and practice regarding folic acid supplementation and neural tube defects. Descriptive and simple linear regression analyses were conducted using SPSS v.26 (SPSS Inc., Chicago, IL, USA). Results: A total of 613 women have completed the questionnaire, from which the majority (46.7%) were aged between 36 and 40 years. About 94% of women heard about folic acid and 80% indicated that its deficiency has some relation to neural tube defects. Approximately 37%, 25.3%, and 23.2% of women reported the proper time for folic acid intake to be during first trimester of pregnancy, before pregnancy, or throughout pregnancy, respectively. Linear regression analysis revealed that increase age and education were significantly correlated with a decrease in folic acid administration (*p* = 0.008) and (*p* = 0.001), respectively. However, there was no association between time of folic acid administration and income or number of parities. Conclusion: Despite the acceptable level of awareness about the relation of folic acid and neural tube defects, our results revealed that more education is required towards the proper time of supplementation among Saudi childbearing women.

## 1. Introduction

Neural tube defects (NTDs) are congenital abnormalities that lead to infant death and serious disability. They appear during embryonic development (typically between the 23rd and 27th day of fetal life) [1,2]. There are two types of NTDs, namely anencephaly and spina bifida. Both NTDs are serious anomalies. A baby with anencephaly may not survive and dies before or soon after birth, whereas a baby with spina bifida can survive, but will have serious functional and health issues and may be mentally retarded [2]. Globally, NTDs influence 323,904 babies [1] and cause 88,000 deaths annually. This could lead to 8.6 million individuals with disabilities [1]. NTDs are the reason behind 29% of newborn deaths in developed countries [3]. Globally, the incidence rate of NTDs is 1 per 1000 births, while the prevalence ranges from 0.5 to 2 per 1000 births in countries without folic acid (FA) supplementation [1,4]. From Riyadh, Saudi Arabia, a study published in 2014 using data from 1996 to 2009 showed that the incidence of NTDs was 1.2 per 1000 live births [5]. Another study from Asir region reported lower incidence rate of NTDs (0.78 per 1000 live births) [6]. An additional study from Jeddah reported only 718 cases on NTDs (including spina bifida and hydrocephalus) registered between 2000 and 2012 [7].

Neural tube defects are caused by a combination of genetic and environmental factors, including maternal nutrition. Countless data indicates that up to 70% of affected births can be prevented by consumption of enough FA before and during pregnancy [8,9]. Over the last two decades, a substantial number of international trials and case-control studies have corroborated this assertion [9]. In a large intervention trial in China, a relatively low dose of FA supplementation (0.4 mg) was effective in protecting against NTDs [10]. Other studies showed that a daily intake of 400 μg of FA before and during the first trimester of pregnancy is approved to prevent NTDs [2,7]. Neural tube defects occur through days 22–28 of fetal growth (before most women know that they are pregnant). Therefore, starting FA intake after the first month of pregnancy is relatively late to prevent NTDs. For this reason, the CDC advises all American women of reproductive age planning to get pregnant to take 400 μg of FA daily [9]. To our knowledge, a few studies have been published from Saudi Arabia aimed at assessing knowledge and practice about FA intake amongst Saudi women of reproductive age [2,11]. According to these studies, women of reproductive age as well as pregnant women have a high level of awareness about FA [2,11]. A more recent study published in 2020 and assessing the knowledge and awareness of Saudi pregnant women about FA intake was found to be in agreement with the previous studies, in which women were found to be aware of the necessity of FA during pregnancy [12]. An additional study conducted in the Hail Region, Saudi Arabia, on three hundred married and childbearing women assessing the awareness and the timing of FA intake revealed that 91.0% of them were aware of FA, 81.0% already knew that FA can prevent NTDs, and 84.0% administered FA prior to or during the first trimester of pregnancy [13]. However, results from these studies cannot be generalized to all childbearing women in Saudi Arabia due to multiple reasons such as low sample size, difference in demographic factors of participants in the region where the studies were conducted, and some studies having been conducted on pregnant women only without considering those of reproductive age (as well as those intending conceive). In this study, we aimed to investigate the knowledge and attitude about FA supplementation among Saudi women of childbearing age whether they were pregnant or not. In addition, we aimed to investigate the source of education from which they learned about FA administration.

## 2. Materials and Methods

A cross-sectional study was conducted using an anonymous online survey to examine the knowledge and practice of Saudi women at childbearing age about FA supplementation. The institutional review board (IRB) reviewed all of the study questionnaire and protocol and approval was obtained from the college of medicine at King Saud University (IRB-E-20/2021). The survey lasted three months from July to the end of September 2021. Women aged >18 years or at childbearing age, who were married, living in Saudi Arabia were included in the study, while psychologically or mentally ill patients were excluded. Before data collection, informed consent was obtained from patients, which confirmed that their data would be kept confidential and used exclusively for research purposes.

### 2.1. Sample Size Calculation

The sample size for this study was calculated using the following formula based on previous FA supplementation prevalence (89%) in Saudi Arabia [12].
*N* = z^2^ × *p* × q/d^2^
where *n* is the sample size, z is the standard normal deviation of 1.96 corresponding to the 95% confidence interval, *p* is the expected prevalence in the proportion of one, q is (1 − *p*), and d is the precision in the proportion of one. If 5% d = 0.05:*n* = (1.96)^2^ × 0.89 (1 − 0.89)/(0.05)^2^
*n* = 150

### 2.2. Design of the Questionnaire

The questionnaire used for this study was prepared based on previous studies published in a similar context [12,14]. The study questionnaire contained two sets of questions. It included both open-ended and closed-ended questions. The first set of questions obtained patient demographic characteristics including age, working status, and educational qualification. Additionally, various variables including the number of pregnancies, abortions, and the number of children with birth defects were assessed. The second set of questions focused on knowledge with a total of eight items with multiple choices. This set included, but not limited to time of intake during pregnancy months, foods that contain FA, and resources of knowledge about FA. The questionnaire was translated to the Arabic language by an independent professional translator, and experts in the field were requested to note independently the suitability of the questions to assess the validity of the questionnaire. After that, the questionnaire was validated using randomly selected 10 women according to inclusion criteria. The reliability was determined using Cronbach’s alpha which was calculated to be 0.79.

### 2.3. Statistical Analysis

Categorical data are presented as frequency and percentage. Continuous variables are presented as mean and standard deviation. The association between demographic data and folic acid intake was assessed using simple linear regression. The *p*-value ˂ 0.05 was considered statistically significant. The data were analyzed using Statistical Package for Social Sciences version 26.0 (SPSS Inc., Chicago, IL, USA).

## 3. Results

A total of 613 women in their childbearing age completed the questionnaire. Approximately half of the participants aged between 36 and 40 years (46.7%), more than the half had one to five pregnancies (52.7%), and majority had a high level of education (universities graduates). Table 1 summarizes other demographic characteristics of the participants. 

Ninety-four percent (94%) of women had heard about FA, and 80% indicated that FA deficiency has a relationship with NTDs, as shown in Table 2. However, only 73.2% were aware of its importance. Regarding the proper time of FA supplementation, 36.9% indicated that they should take it during the first trimester of pregnancy, while 25.3% said that it should be taken before pregnancy, and 23.2 % believed it should be taken during all pregnancy’s trimesters. Regarding the practice of FA intake during pregnancy, 88.3% reported taking FA supplementation, while surprisingly 11.7% indicated they didn’t take it. In addition, and during the pregnancy, 47.8% of participants stated that they consumed leafy green vegetables, 45.6% did not consume too much, and only 6.7% did not consume any. More information about knowledge and practice of women about FA is provided in Table 2.

As for the source of knowledge about FA intake, most of women obtained their information from health care providers (*n* = 416; 67.9%), followed by media (*n* = 97; 15.8%), books and magazines (*n* = 36; 5.9%), friends or relatives (*n* = 34; 5.5%), and other resources (*n* = 36; 5.9%), as shown in Figure 1.

To determine the relationship between appropriate time to take FA, age, income, education level, and several parities, a multiple regression linear model was utilized in which age, income, education level, and several other parities were considered as explanatory variables and time to FA was taken as the dependent variable. As the age and education of women increase, there was a significant decrease in the intake of FA during the first trimester ((*p* = 0.008) and (*p* = 0.001), respectively, as shown in Table 3). However, there was no association between time of FA intake and income or number of parities, shown in Table 3.

## 4. Discussion

The use of folic acid (FA) supplementation by pregnant women or those who are planning to conceive has been recommended by the United States Public Health Service, the Institute of Safe Medicine (IOS), and the Center for Disease Control and Prevention (CDC), since it has a preventative effect on birth defects, such as NTDs, and to treat FA deficiency anemia which helps women as well as fetus to avoid blood pathologies [15,16,17]. It is common knowledge that a well-balanced diet rich in carbs, vitamins, proteins, and minerals during or before conceiving is essential for womb development and healthy pregnancy [2,12,18]. However additional supplementation of various micronutrients, such as FA, will fill in any nutritional shortage that may arise during pregnancy.

In this study, the majority of the women had heard (94%) and were aware of FA requirements, especially during pregnancy (which is similar to the previous studies by Alreshidi et al. and Al-Holy et al. among Saudi women) [13,19]. The proportion is more than an earlier study published by Alfadly and colleagues on Yemeni pregnant women, among which 62% were aware of FA need [14]. In this study, the importance of FA supplementation was reported by 73.2% of the Saudi women, indicating that most of the women were knowledgeable about the importance of FA. Our findings are similar to the results of the study conducted in Riyadh, Saudi Arabia among pregnant women attending clinics, in which 80% of the interviewed women were aware of the benefits of taking FA supplements [20]. Similarly, a case-control study of Saudi mothers receiving prenatal care at a major hospital in Riyadh found that 93% took FA [21]. Additionally, it has been shown that the majority of Saudi pregnant women had a good background about FA supplementation in spite of the low percentage of them knowing the importance of FA supplementation during pregnancy (42.2%) [19]. Intriguingly, however, a previous study from the Almadinah Almunawwarah region of Saudi Arabia reported unfavorable results where women of reproductive age attending university outpatient’s clinic had fairly little knowledge and attitude towards FA intake [22]. These findings highlight that the level of knowledge and attitude towards FA consumption among Saudi women of childbearing age vary based on their region. Thus, the amount of effort to improve FA consumption and knowledge among this category of the population is potentially influenced by their region.

Many previous studies reported the importance of FA intake before or during pregnancy among childbearing-aged women [23,24,25,26]. Additionally, studies also showed that having adequate knowledge about FA consumption is essential to prevent birth defects among pregnant women [25,26,27]. In our study, the majority of Saudi women (80.3%) believed that FA deficiency has a relationship with NTDs, which is consistent with previous studies by Al-Holy et al. and Alreshidi et al. among Saudi women, who reported that between 81% and 80.1% of the women thought FA deficiency causes neural abnormalities [13,19]. Nevertheless, although the current finding showed that majority of Saudi women consumed FA, they in fact did not know the proper time of administration (such that 25.3% reported to take it before pregnancy, while one-third believed to take it during the first trimester, and only 23.2% believed that it should be administered during all stages of pregnancy). According to the previous study by Al-Holy and colleagues, only 10% of those who claimed to know the proper time for FA supplementation indicated that it should be taken before and during the first trimester of pregnancy [13]. Similarly, Al Hossain et al. discovered that 35% of Emirati women used FA during the first month of pregnancy, 29.5% used FA before the last menstrual cycle, and 9.6% of the women had no idea when FA should be administered [27].

Our study revealed health care providers as the source of information about FA among majority of women (67.9%) which was similar to the previous study by AlDuraibi and Al-Mutawa among Saudi pregnant women in 2020 [12] and Kamau et al. among pregnant women in Kiambu County, Kenya in 2019 [17], Al-Holy et al. among childbearing age women in Saudi Arabia [18]. Similarly, another recent study by Alreshidi et al. in 2018 found that most Saudi women gained their knowledge about FA from their doctors and nurses [18]. Although Alfadly et al. reported that 39.5% of Yemeni childbearing women obtained that from friends or relatives, 20.4% found it from a physician [14].

Participants’ age and education in our study were found to be significant predictors for FA intake (during the first trimester), which was similar to a previous study among Saudi women by Al-Holy et al. and Egyptian women by Al-Darzi et al. [13,28]. Although these studies showed university-educated women had significantly higher use of FA supplementation than those with lower education, our findings in contrast demonstrated the opposite where higher education was associated with lower FA consumption. It is clear that all university educated women in our study consumed during their pregnancy natural resources of FA such as fruits and green leafy vegetables (as university educated women represented 86% of participants and those who consumed natural FA resources were 93.4%), which potentially made them perceive less need to FA supplementation than those with lower education. Interestingly, a study investigating the awareness and intake of FA among Lebanese women at childbearing age showed that although higher education was associated with more awareness, it had no impact on the FA intake [29]. Similar results were reported by Macwalter et al. on Saudi pregnant women where no association was found between level of education and folic acid intake [30]. Therefore, higher education is not always correlated with an increase in FA supplementation. To complicate this more, a recently published survey-based study by AL-Mohaithef et al. on university-educated females from the Saudi Electronic University in Jeddah, Saudi Arabia showed low proportion were using FA supplement (47.1%) [31]. This underlines an additional potential factor that cannot be ignored despite its lower applicability to our findings that is the field of education. In other words, although women in the aforementioned study [31] were university-educated, being highly educated in fields other than healthcare may not translate into an optimal consumption of FA and other supplements before or during pregnancy. Although this might have influenced our results, we could not capture it by the utilized questionnaire. Collectively, this reiterates the notion that use of FA supplement might potentially be impacted by the education field beside the education level of women at childbearing age. With respect to age, we found that consumption of FA corelated negatively with increased women’s age (*p* = 0.008). In other words, increasing age was associated with less FA intake which contradicts the results previously reported on American [32,33], Ethiopian [34] and Lebanese [35] childbearing women. However, our findings are supported by another study conducted on Lebanese childbearing women where more FA users were found among younger compared to older women [31]. Additionally, although Macwalter and colleagues showed no impact of age on starting FA intake before pregnancy among Saudi pregnant women when those aged <35 years compared to women >35 years, the proportion of non-users during pregnancy in later was higher compared to the former group, recapitulating the observation demonstrated by our study [30].

## 5. Conclusions

Our study confirmed the increased awareness of Saudi childbearing about FA consumption and its benefit as a preventive for neural tube defects during pregnancy. Nevertheless, a lack of knowledge exists on the proper time of FA administration, which is very crucial. Additionally, the impact of education and age on FA intake among childbearing women in Saudi Arabia deserves a revisit. Overall, more efforts should be made by healthcare professionals such as physicians and pharmacists to educate women of reproductive age about the importance of timing of FA supplementation in preventing pregnancy-related disorders.

## Figures and Tables

**Figure 1 nutrients-14-00357-f001:**
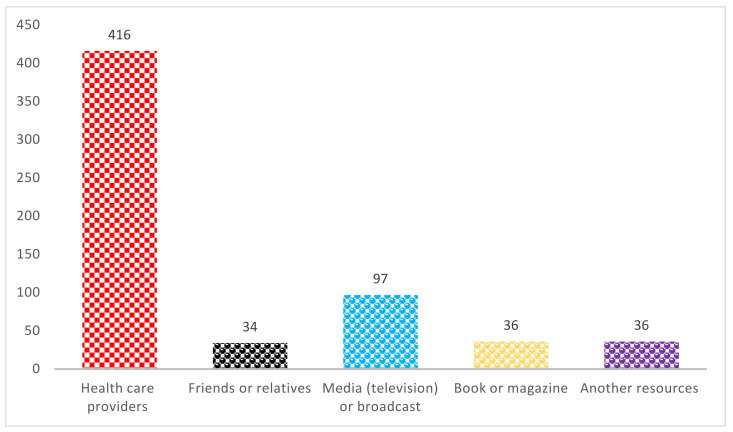
Source of knowledge of FA among participants.

**Table 1 nutrients-14-00357-t001:** Demographic characteristics of participants (*n* = 613).

Variables	*n* (%)
Age groups	
16–20	16 (2.6)
21–25	25 (4.1)
26–30	107 (17.5)
31–35	179 (29.2)
36–40	286 (46.7)
Income	
Meets family needs	457 (74.6)
Less than family needs	71 (11.6)
Exceeds family needs	85 (13.9)
Education level	
Illiterate	5 (0.8)
Primary school	6 (1.0)
Secondary school	75 (12.2)
University	527 (86.0)
Number of Parity	
1–5	491 (80.1)
6–10	116 (18.9)
11–15	6 (1.0)
Number of abortions	
None	479 (78.1)
1–3	124 (20.2)
4–7	10 (1.6)
Number of anomalies	
Non	563 (92.0)
One episode	45 (7.0)
Two episodes	5 (0.8)

**Table 2 nutrients-14-00357-t002:** Knowledge and practice of pregnant women about FA (*n* = 613).

Variables	*n*	%
Heard about FA (Yes)	575	93.8
Folic acid is important	449	73.2
When to supplement with folic acid		
Before pregnancy	155	25.3
During the first trimester	226	36.9
During the second trimester	5	0.8
Last trimester	1	0.2
Through all stages	142	23.2
Does not know	84	13.7
Thinks that folic acid deficiency has a relationship with neural tube defect (True)	492	80.3
The practice of pregnant women towards folic acid		
Take folic acid (Yes)	541	88.3
During pregnancy, consume vegetables, fruits, and green leaves		
Yes	293	47.8
No	41	6.7
Not too much	279	45.6

**Table 3 nutrients-14-00357-t003:** Results of FA intake during the first three months of pregnancy with some demographic features of women using regression analysis.

Variables	B	Std. Error	t	*p*-Value
Age	−0.206	0.078	−2.64	0.008
income	−0.163	0.104	−1.56	0.119
Education level	−0.934	0.169	−5.5	0.001
Number of Parity	0.179	0.189	0.948	0.948

## Data Availability

The datasets used and analyzed during the current study are available from the corresponding author on reasonable request.

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
