# Peer review of "Knowledge and Practice of Childbearing Women in Saudi Arabia towards Folic Acid Supplement—Evidence from a Cross-Sectional Study"

_nutrients, 2022, doi:10.3390/nu14020357_

Round 1
Reviewer 1 Report
This article is sufficiently revised following my previous peer-reviewed opinion.
Author Response
Response 1: The manuscript did significantly improve after considering your comments. We appreciate all your time and effort dedicated to that.
Reviewer 2 Report
This article evaluates the knowledge and practice of Saudi Arabian women regarding folic acid supplementation. Overall, the study is well conducted with a good sample size, and the manuscript is written well. I have the following minor concerns/questions.
Line 130 - "94% reported taking FA supplementation" - In Table 2 shows 541 women (88.3) taking folic acid - Clarify this.
Line 245 - "Neuronal tube defects" - would recommend using the same term "neural" at all places for consistency.
Author Response
Comment 1: This article evaluates the knowledge and practice of Saudi Arabian women regarding folic acid supplementation. Overall, the study is well conducted with a good sample size, and the manuscript is written well. I have the following minor concerns/questions.
Response 1: We appreciate the reviewer for time and effort dedicated to review our manuscript. Thank you.
Comment 2: Line 130 - "94% reported taking FA supplementation" - In Table 2 shows 541 women (88.3) taking folic acid - Clarify this
Response 2: We appreciate your comment. The percentage of women taking FA supplement during pregnancy has been corrected and changed to 88.3 % taking into consideration all the whole sample of the study. This has been tracked in the manuscript.
Comment 3: Line 245 - "Neuronal tube defects" - would recommend using the same term "neural" at all places for consistency.
Response 3: Thanks for your suggestion. It has been changed to “neural” for consistency.
This manuscript is a resubmission of an earlier submission. The following is a list of the peer review reports and author responses from that submission.
Round 1
Reviewer 1 Report
In this article, autors evaluate the knowledge and practice taking folic acid supplements in Saudi women and showed that the recognition is spreading.
As a comment, I will state the followings. 
1)You have shown the interesting and unexpected finding.
L-22: ”linear regression analysis revealed that an increase age and education of women was significantly correlated with a decrease in the folic acid uptake.
L-140 : “As the age and education of women increase ,there was a significant decrease in the uptake of FA”
L-202 : “Lower FA consumption was correlated to higher education and higher age”
This finding is different from the results of conventional surveys. This research result is not convincing to many researcher including me. However, the research results are statistically clear. So it is especially important to give a convincing explanation in the discussion.
2) You discribe the amount of FA intake as followingâ‘ and â‘¡
① L-50 relatively low dose (0.4 mg )
② L-51: sufficient intake of FA
Please describe the amount of FA instead of “Sufficient FA”, like L-50..
“Sufficienrt” is an unclear expression and should not be used.
3) Table 2 shows 88.3% of people taking FA. Quite a lot of women are taking FA. Therefore, please show the incidence of spina bifida in the researching area, being expected that the incidence of NTDs is fairly low.
5)L-152~L-153 about “To boost iron or heme levels”,
Usually blood iron and heme do not increase unless iron is administered. However, in the case of folic acid deficient anemia, the anemia could be expected to improve by administration of folic acid. Therefore this expression should be corrected.
ï¼–) Correct the following expressions
① L-39 “worldwide”
Rewrite “worldwide” to “worldwidely”
② L-156 “although”
“Although” should be removed.
➂ L-164 a study conducted in Saudi Arabia。
Rewrite “a study conducted in Saudi Arabia” to “the study conducted in Saudi Arabia”.
④ l-169 “ a good level of knowledge about FA supplementation”
Rewrite “a good level of knowledge about FA supplementation” into a clear expression. This is difficult to be understood.
⑤ l-179   “a previous study by Alreshidiaet al.”
Rewrite “a previous study by Alreshidiaet al.” to “the previous study by Alreshidiaet al.”
Author Response
We thank the reviewer for his comments. All raised points have been addressed.

Reviewer 2 Report
Figure 1 is useless as the data are presented in the Results
Tables 2 and 3 are not quoted anywhere
Also, the methodology needs to be improved since there are three very basic surveys currently
Author Response

(The authors gave the same response as above.)
